# A general patterning approach by manipulating the evolution of two-dimensional liquid foams

Zhandong Huang[1,2], Meng Su[1,2], Qiang Yang[1,2], Zheng Li[1,2], Shuoran Chen[3], Yifan Li[1,2], Xue Zhou[1,2], Fengyu Li[1] & Yanlin Song[1]

The evolution of gas-liquid foams has been an attractive topic for more than half a century. However, it remains a challenge to manipulate the evolution of foams, which restricts the development of porous materials with excellent mechanical, thermal, catalytic, electrical or acoustic properties. Here we report a strategy to manipulate the evolution of two-dimensional (2D) liquid foams with a micropatterned surface. We demonstrate that 2D liquid foams can evolve beyond Ostwald ripening (large bubbles always consuming smaller ones). By varying the arrangement of pillars on the surface, we have prepared various patterns of foams in which the size, shape and position of the bubbles can be precisely controlled. Furthermore, these patterned bubbles can serve as a template for the assembly of functional materials, such as nanoparticles and conductive polymers, into desired 2D networks with nanoscale resolution. This methodology provides new insights in controlling curvature-driven evolution and opens a general route for the assembly of functional materials.

[1] Key Laboratory of Green Printing, Institute of Chemistry, Chinese Academy of Sciences (ICCAS), Beijing Engineering Research Center of Nanomaterials for Green Printing Technology, Beijing National Laboratory for Molecular Sciences (BNLMS), Beijing 100190, China. [2] University of Chinese Academy of Sciences, Beijing 100049, China. [3] Jiangsu Key Laboratory for Environmental Functional Materials, Research Center for Green Printing Nanophotonic Materials, Institute of Chemistry, Biology and Materials Engineering, Suzhou University of Science and Technology, Suzhou 215009, China. Correspondence and requests for materials should be addressed to Y.S. (email: ylsong@iccas.ac.cn).

Ostwald ripening or coarsening is prevalent in the evolution of many systems, such as gas–liquid foam[1], droplets in a breath figure[2], particles in a supersaturated solution[3] and ubiquitous cellular structures in meal, ceramics and so on[4]. In these systems, big particles always consume adjacent smaller ones. Breaking the Ostwald ripening to manipulate the evolution of these systems will be of great significance in many subjects, such as the physics of foams, metallurgy, topology, morphology, biology and so on[5–7]. With studies conducted for over half a century, the evolution of gas–liquid foams has been well understood[8–11]. Some elegant theories have been developed to describe the evolution of foams, for example, Lemlich's model for evolution of fairly wet gas–liquid foams[8], von Neumann's law for the coarsening of 2D dry foams[9] and other studies modelling 3D evolving foams[10,11]. To control the evolution of foams, many strategies have been proposed, such as using an external field of temperature[12] or magnetism[13], foams stabilizers of nanoparticles[14,15], proteins[16] or surfactants[17,18], insoluble fluorated gases[19] and so on. However, most of these methods can only slow or stop the evolution/aging of foams by weakening the gas diffusion between bubbles. Therefore, manipulating the evolution of 2D liquid foams beyond the Ostwald ripening remains a significant challenge.

Recently, the micro/nanoscale structural features of a surface were believed to create a gradient in Laplace pressure or saturated vapour pressure, which encourages fog collection[20], water drop directional transport[21,22] or condensation[23] and altering of the morphology of liquid crystals[24]. In this study, we demonstrate that a micropatterned surface can be used to manipulate the evolution of 2D liquid foams. The pillars on a micropatterned surface can effectively control the radius of the curvature of the bubbles in foams, thus mediating gas diffusion between bubbles. In controlled evolution, 2D liquid foams are endowed with a reverse Ostwald ripening mechanism that forbids the large bubbles from growing infinitely. In addition, when the volume fraction of gas is insufficient, and the gas is distributed uniformly, the evolution exhibits the gathering effect, that is, the bubbles selectively evolve towards domains, where they have a larger radius of curvature. This fundamental understanding enables us to prepare various 2D bubble patterns. Furthermore, with these patterned bubbles as a template, an efficient, clean and sustainable strategy is developed for assembling functional materials into desired 2D networks with a nanoscale resolution for the fabrication of high-precision electronic devices. It is expected that our strategy will further the understanding in controlling curvature-driven evolution and provides a general method for the 2D assembly of functional materials.

## Results

### Evolving the 2D liquid foams into hexagonal bubble arrays.
In foams, drainage from gravity, coalescence and coarsening from inter-bubble gas diffusion mainly affect the evolution[6]. However, for foams in 2D, the drainage from gravity is negligible, and coalescence between bubbles can be easily avoided by adding surfactants[25]. Therefore, inter-bubble gas diffusion caused by differences in Laplace pressure between bubbles dominates the evolution. To control this process, we have designed a pillar-structured silicon substrate, as shown in Fig. 1a. The arrangement and interval of the pillars vary with requirement, and the radius and height are fixed at 5 and 20 μm, respectively. In the experiments, pillars act as discrete points for interacting with growing bubbles and gaps between the pillars allow gas exchange among bubbles. We prepared the 2D foams by a decomposition reaction, that is, the decomposition of urea

peroxide catalysed by Ag nanoparticles (AgNPs) (Supplementary Fig. 1). The foams produced in the confined 2D space consisting of a glass coverslip and the patterned surface (Fig. 1b). Foams with gas volume fraction from 0% to 94% could be obtained by varying concentration of AgNPs and urea peroxide (Supplementary Fig. 2). In this work, we mainly focus on the evolution of wet foams or bubbly liquid in 2D (ref. 6), where most of the bubbles maintain their roundness at the initial state of the evolution.

When bubbles were produced in the confined space, they would be corralled into polygonal cells consisting of discrete micropillars (Supplementary Fig. 1e), and their evolution would be regulated (Supplementary Fig. 1f). We started the investigation by evolving the 2D foams into hexagonal bubble arrays. The formation of hexagonal bubble arrays is shown in Fig. 1c and the first part in Supplementary Movie 1. Initially, large bubbles obeyed the Ostwald ripening mechanism to consume smaller neighboring ones. Once large bubbles filled the hexagon cells in which they located, they were arrested and stopped growing, with other bubbles continuing to evolve. Afterward, each hexagonal cell was filled with a round or deformed bubble. Meanwhile, because of evaporation from the side edges, drainage occurred, and the boundaries of the bubbles merged into liquid films, eventually forming the hexagonal bubble arrays. In the evolution of 2D foams without micropillars on the surface (the second part in Supplementary Movie 1), large bubbles consumed the smaller ones until all of them disappeared, following the common Ostwald ripening.

To explain the phenomenon mentioned above, we investigated the interaction between bubbles and micropillars by magnifying the gas–liquid interface during evolution (Fig. 1d; Supplementary Fig. 3). As shown in Supplementary Fig. 3, when a bubble grows in the hexagonal cell and meets with the pillars, its boundary will deform into several meniscuses. Essentially, the deformation caused by the surface tension obeys the minimal energy principle, and the deformed shape has an equal radius of curvature[7]. When the bubble grows in the hexagonal cell consisting of discrete micropillars, its maximum radius of curvature is limited. Once the bubble reaches the maximum circular size, any increase in gas volume from diffusion will result in the formation of meniscuses, which have a small radius of curvature. The high Laplace pressure in the meniscuses allows the smaller bubbles to absorb gas from the meniscuses. Therefore the deformed bubble cannot grow further. During the evolution, small bubbles shrink gradually until vanishing, while large bubbles grow in the following three states: free growth, exactly filling the hexagonal cell and deformation, as shown in Fig. 1d. Once a bubble reaches the maximum circular size and meets the pillars, the large radius of curvature ($R_2$) enables it to grow. The boundary of the bubble then deforms into several meniscuses under the action of the pillars. The deformed bubble has a small radius of curvature ($R_3$), which allows it to be absorbed by the surrounding smaller bubbles, for example the bubble with a radius of $R_1$ shown in Fig. 1d ($R_1 > R_3$). In this study, the phenomenon of a large deformed bubble being adsorbed by the adjacent smaller bubble is called the reverse Ostwald ripening. It suppresses the growth of large bubbles and encourages the formation of arrays of equal-sized hexagonal bubbles.

### The pillar interval determines the reverse Ostwald ripening.
When the pillar interval increased from 35 to 60 μm, some defects formed (Supplementary Fig. 4), and the reverse Ostwald ripening and defect formation coexisted in the evolution (Fig. 2a,b). The transfer of gas from one bubble to a nearby

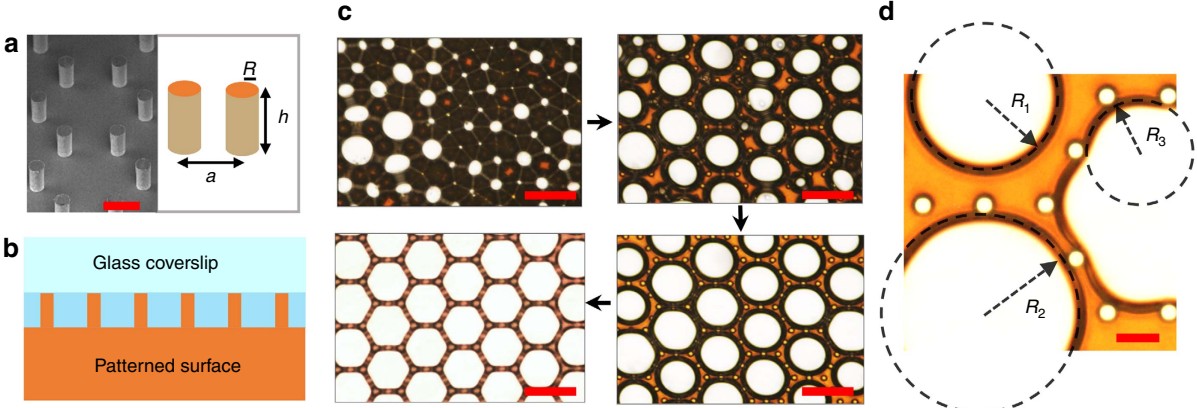

**Figure 1 | Evolving 2D foams into hexagonal bubble arrays.** (**a**) Scanning electron micrograph (SEM) of the patterned surface and schematic diagram for showing the parameters of the pillars: the radius, $R$, the height, $h$ and the central distance of adjacent pillars, $a$. (**b**) Side view of confined 2D space consisting of a flat glass coverslip and the patterned surface. (**c**) A video recording from a top view was made of the formation of hexagonal bubble arrays (Supplementary Movie 1 for complete evolution). (**d**) Magnified image of the gas-liquid interface during the evolution to illustrate the effect of the pillars on the evolution of 2D foams. Bubbles with radii of curvature of $R_1$, $R_2$ and $R_3$, show three states of growing bubble: free growth, exactly filling the hexagonal cell (reaching the hexagonal cell's maximum circle size) and deformation. Scale bars, **a** 20 μm, **c** 100 μm, **d** 30 μm.

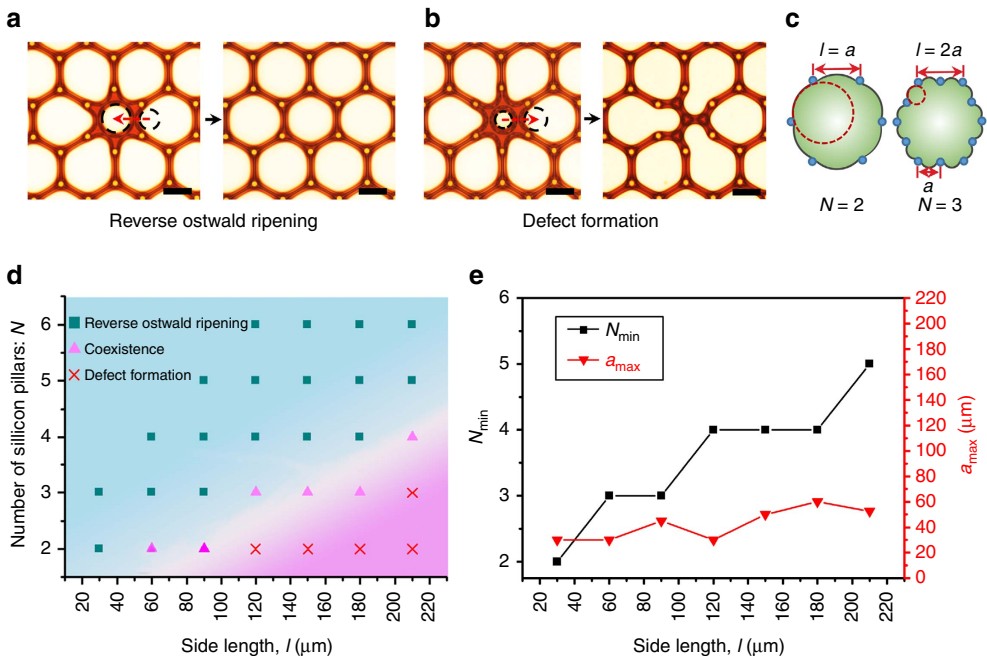

**Figure 2 | The influence of pillar interval on evolution of 2D foams.** (**a**,**b**) The phenomena of reverse Ostwald ripening (the central smaller bubble consuming surrounding larger bubbles in **a**) and defect formation (the central smaller bubble being consumed by surrounding larger bubbles in **b**) coexist during the evolution when increasing the pillar interval from 35 to 60 μm. The longer series of images corresponding to **a** is shown in Supplementary Fig. 5. The red dashed arrows in **a**,**b** denote the gas transfer direction among the central bubble and surrounding bubbles due to difference in radius of curvature (denoted with black dashed circle). (**c**) Illustration of bubbles deforming in hexagonal cells with the same $a$ and different $n$ values. The illustrated bubbles in **c** have the same area. This suggests that when more pillars are present on each side, the bubble will have a smaller radius of curvature. (**d**) Dependencies map of the preparation of bubble arrays on the side length ($l$) and the number of silicon pillars in each side ($N$). The Reverse Ostwald ripening (donated as a square) corresponds to the preparation of perfect hexagonal bubble arrays; the coexistence (donated as a triangle) of Reverse Ostwald ripening and defect formation leads to partly successful preparation of hexagonal bubble arrays; defect formation (donated as a cross) is unable to form bubble arrays. The significance of $N$ and $l$ is illustrated in **c**. (**e**) The dependency of $N_{min}$ (the least number of silicon pillars on each side that it requires for preparing perfect bubble arrays) and the corresponding $a_{max}$ (the maximum pillar interval) on the side length $l$. Scale bars, **a**,**b** 50 μm.

bubble depends on the local radius of curvature of each bubble. As shown in Fig. 2a, the central small bubble has a larger local radius of curvature than that of the surrounding deformed large bubbles. Thus, gas diffusion occurs from the larger bubbles to the small bubble. The small bubble grows gradually and the larger bubbles shrink during the evolution (Supplementary Fig. 5), eventually forming arrays of equal-sized bubbles. Conversely, as shown in Fig. 2b, the radius of

curvature of the spherical bubble at the center is smaller than that of the larger bubbles, hence the diffusion occurs from the smaller bubble to the larger bubble, finally forming a defect. The pillar interval is a key factor because it limits the minimum radius of curvature for the meniscuses, which is $a/2$ as calculated in Supplementary Note 1. Once the radius of curvature of the deformed large bubble reaches its minimum, the surrounding bubbles with radii above $a/2$ will grow, showing the reverse Ostwald ripening, while bubbles with radius below $a/2$ will shrink, forming a defect. According to Lemlich's theory[8], the evolution equation of wet foams can be deduced as

$$\frac{dr}{dt} = K\left(\frac{1}{\rho} - \frac{1}{r}\right) \tag{1}$$

where $K$ is positive and can be viewed as a constant, $t$ is time. $r$ is the radius of the bubbles, and $\rho$ is instantaneous mean radius. The equation suggests that, any bubbles with a radius larger than $\rho$ will grow and those with a radius smaller than $\rho$ will shrink. The same conclusion is obtained when the equation is applied to 2D wet foams (Supplementary Note 2). If $a/2$ is less than $\rho$, the deformed larger bubbles will shrink once their radii of curvature reach the minimum radius of curvature, $a/2$. Thus, the reverse Ostwald ripening will always happen. Since $\rho$ gradually increases during the evolution[8], the minimum $\rho$ is at the initial state of the evolution (denoted as $\rho_0$). Therefore keeping a condition in which $a/2 < \rho_0$ can effectively forbid defect formation. The discussion above can be applied to any other polygonal cells, and a detailed explanation is given in Supplementary Note 3.

To prepare perfect hexagonal arrays of larger bubbles, increases in the side length of the hexagonal cells result in the increases in the pillar interval, which leads to the formation of defects. An alternative method is to increase the number of silicon pillars on each side (denoted as $N$) when increase the side length (denoted as $l$). As shown in Fig. 2c, the pillar interval $a$ equals $l/(N-1)$. For the same side length $l$, more pillars on each side $N$ correspond to a smaller the pillar interval $a$, and an increased occurrence of the reverse Ostwald ripening phenomenon. This deduction agrees well with the experiments exhibited in Fig. 2d. The squares on the map mean that $N$ is large and $a/2$ will be small and less than $\rho$; thus, only reverse Ostwald ripening happens, eventually forming perfect hexagonal bubble arrays. The crosses in the map suggest that $N$ is small and $a/2$ will be much larger than $\rho$; thus, only the defect formation will happen. The triangles show that if $N$ is not too large, $a/2$ will be slightly more than $\rho$, and some bubbles shows the reverse Ostwald ripening (Fig. 2a) and others exhibits the defect formation (Fig. 2b). The coexistence of reverse Ostwald ripening and defect formation leads to the formation of partly hexagonal bubble arrays (Supplementary Fig. 4b). The least number of silicon pillars on each side and the corresponding maximum pillar interval for preparing perfect bubble arrays of different side length were summarized in Fig. 2e. With the increase of side length, the $N_{min}$ increases significantly for rejecting the increase of pillar interval. Therefore the maximum pillar interval only increases slightly. The allowed maximum pillar interval, $a_{max}$, can be considered $2\rho$ because both are the allowed maximum pillar interval for forming perfect bubble arrays, that is, $a/2 \leq a_{max}/2 \approx \rho$.

**The special arrangement of pillars contributes the gathering effect.** We then changed the arrangement of pillars. The pillars on the micropatterned surface were arranged into three kinds of cells, regular dodecagonal, hexagonal and square cells (Fig. 3a–d) with equal pillar intervals. We found that this designed substrate

evolved the 2D foams into discrete dodecagonal bubble arrays when the gas volume fraction was $<57\%$ (Supplementary Movie 2). Bubbles were produced in all cells at first, while bubbles in the dodecagonal cells adsorbed all the bubbles in the square and hexagonal cells during the evolution. Figuratively speaking, bubbles in the dodecagonal cells have the gathering effect collecting surrounding bubbles. Interestingly, bubbles could fill both dodecagonal and hexagonal cells by increasing the gas volume fraction. Only the bubbles in the square cells were consumed (Supplementary Fig. 6).

To explain this, we calculated the radius of curvature for these bubbles varying with the increase of volume (the area for a 2D bubble), as shown in Supplementary Note 1; Fig. 3e. When bubbles grow in these three kinds of cells, as the area increases, changes in the radius of curvature are similar. They all gradually reach a maximum and then sharply shift back to a minimum of $a/2$ before slowly increasing (Fig. 3e). The difference in the maximum means that the dodecagonal cell allows the growing bubble a larger maximum circle size than the square or hexagonal cell does. When a bubble in a square or hexagonal cell reaches the maximum circle size and starts to deform (indicated by the red and blue arrows in Fig. 3e), the bubble in the dodecagonal cell can grow continuously. In addition, since the bubbles distribute uniformly, more gas will be present in the dodecagonal cells at the initial state of the evolution than that in square or hexagonal cells. The bubbles in dodecagonal cells have not reached the maximum circle size at the initial state since the total gas is not sufficient. With the evolution, bubbles in dodecagonal cells have larger circle sizes (larger radius of curvature) than that in square or hexagonal cells, which causes an unbalanced pressure between the dodecagonal cells and the square or hexagonal cells. Therefore, bubbles in dodecagonal cells will adsorb the surrounding bubbles until filling the dodecagonal cells. When volume fraction of gas is sufficient ($>57\%$), bubbles in the hexagonal cells can survive because the consumption of the bubbles in square cells has filled the dodecagonal cells. Bubbles in hexagonal cells also consume the bubbles in square cells until filling the hexagonal cells. The explanation above can be applied to any other polygonal cells, and a detailed discussion of this is provided in Supplementary Note 3.

**Simulation for reproducing the evolution.** To test the mechanisms above, we next made a simulation to reproduce the evolution. The evolution equation of 2D foams was deduced from applying Lemlich's theory[8] to 2D foams (Supplementary Note 2):

$$\frac{dA}{dt} = KL\left(\frac{1}{\rho} - \frac{1}{r}\right) \tag{2}$$

where $K = RTJ\sigma/P_0$ and is assumed to be constant during the evolution, and $\rho = \sum_{i=1}^{n} L_i / \sum_{i=1}^{n} \frac{L_i}{r_i}$, which denotes the instantaneous mean radius of the foams[8]. The $dA/dt$ is the rate of change of the area of a bubble. $L$, $r$ are the boundary perimeter and radius of curvature for the bubble in 2D, respectively. $n$ is the number of all bubbles in the foams, and $R$ is the gas constant. $T$ is the temperature and $J$ is the effective permeability to the transfer. $\sigma$ is surface tension of the solution and $P_0$ is atmospheric pressure. Relationships between $L$, $A$ and $r$ should be deduced for solving the equation, which can be obtained by studying the effect of pillars on the evolution of 2D foams. The relevant equations are

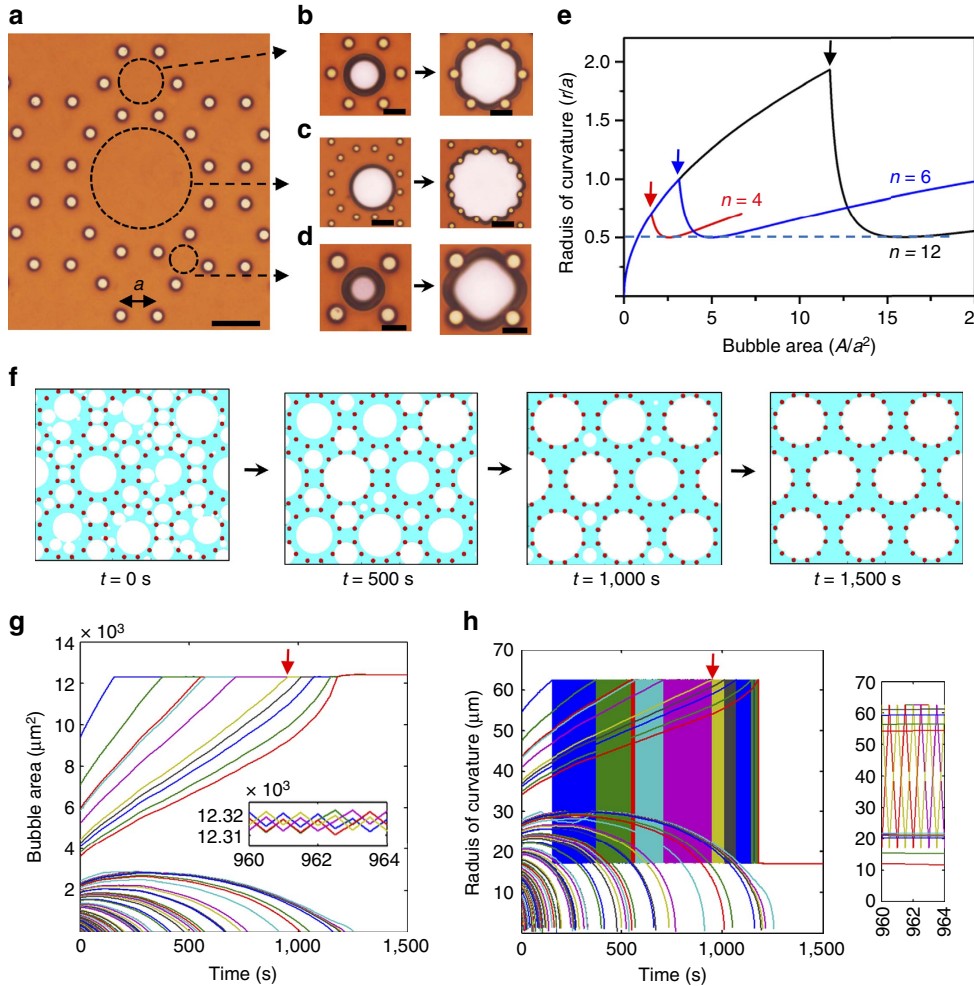

**Figure 3 | Simulations for reproducing the gathering effect. (a)** Pillars on the substrate are present in different arrangements. They divide the 2D space into hexagonal, dodecagonal and square cells, marked by dashed circles. The three kinds of cells have identical side lengths, *a*, as indicated. **(b–d)** Microscope observations of bubble deformation at the corresponding position marked in **a**. **(e)** Quantitative description of the radius of curvature varying with bubble area when bubbles grow and deform in square ($n=4$), hexagonal ($n=6$) and dodecagonal ($n=12$) cell. Bubble area ($A/a^2$) and radius of curvature ($r/a$) are non-dimensionalized. Arrows above the curves denote that bubbles begin to deform from roundness and their radii of curvature decrease sharply. **(f)** The evolution process at different times in the simulation, reproducing the gathering effect (see Supplementary Movie 3 for complete simulation). **(g,h)** Area **(g)** and radius of curvature **(h)** variations of every bubble in the foams as a function of time. In the graphs, each curve denotes the area variation or radius of curvature for a bubble in the simulation in **f**. Insets in **(g,h)** are details with an enlarged scale of the curves marked by the red arrows. They show that, when bubbles fill the dodecagonal cells, a very small change in area can result in a significant change in the radius of curvature. Scales, **a** 50 μm, **b** 20 μm, **c** 50 μm, **d** 20 μm.

(Supplementary Note 1):

$$A = f(r) \atop L = f(r) \left\{ \begin{array}{lll} L = 2\pi r & A = \pi r^2 & A \in \left[0, \frac{\pi a^2}{4\sin^2(\pi/n)}\right] \\ L = n\theta r & A = \left(\frac{a^2}{4\tan(\pi/n)} + \frac{1}{2}\theta r^2 - \frac{a^2}{4\tan(\theta/2)}\right) \times n & A \in \left[\frac{\pi a^2}{4\sin^2(\pi/n)}, \infty\right] \end{array} \right.$$

(3)

where $n = 4$, 6 or 12 meaning that bubbles grow in square, hexagonal or dodecagonal cells, *a* is the distance between centers of adjacent pillars, and $\theta$ is defined in Supplementary Note 1, satisfying $\sin(\theta/2) = a/2r$.

Combining the equations of (2) and (3), we carried out the simulation, and detailed information is provided in the Methods section. In the simulation (Fig. 3f; Supplementary Movie 3), areas of small bubbles gradually reduce to zero, while areas of large bubbles cannot infinitely increase. The explanation (Fig. 3g,h) is that, when a bubble exactly fills a dodecagon cell, a small increase in area (inset in Fig. 3g) will

reduce the surface curvature radius from the maximum to the minimum (inset in Fig. 3h), thus reducing the growing ability to the weakest. This allows it to be adsorbed by surrounding smaller bubbles, and the large deformed bubble will shrink back to the dodecagonal cell until having a large radius of curvature. This reverse Ostwald ripening mechanism ensures that polygonal cells can effectively trap bubbles by forbidding them from overstepping the polygonal cells. In addition, foams with various gas volume fractions were also simulated (Supplementary Fig. 7), which agreed well with the gathering effect.

**Various bubble patterns**. The narrow space between pillars endows the evolution of 2D foams with reverse Ostwald ripening mechanism for arresting bubble growth. In addition, the special arrangement of pillars allows the evolution exhibit a gathering effect to localize bubbles. Based on this fundamental

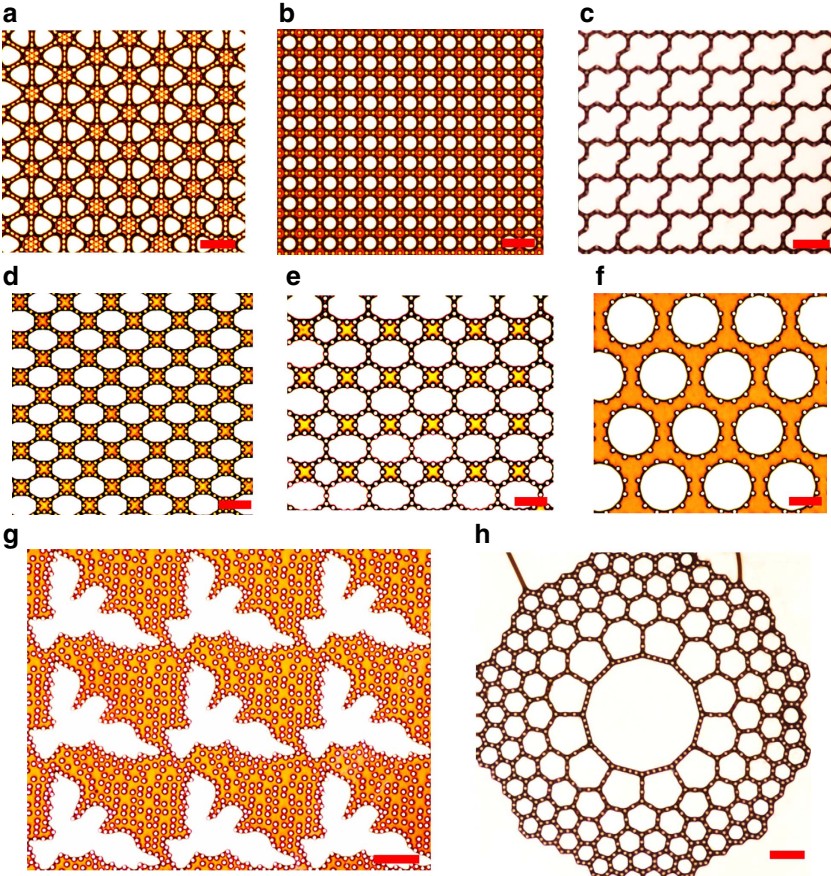

**Figure 4 | Various bubble patterns.** (**a**) Triangular bubble arrays. (**b**) Square arrayed round bubbles. (**c**) Cross-shaped bubble arrays. (**d**) Octagonal bubble arrays. (**e**) Bubble patterning consisting of hexagon and octagon. (**f**) Discrete dodecagonal bubble arrays. (**g**) Peace-dove-shaped bubble arrays. (**h**) Conformal pattern of a hexagonal bubble arrays with rotational symmetry[26], consisting of a dodecagonal bubble and hexagonal bubbles of different size. All scale bars, 100 μm.

understanding, we evolved 2D foams into various patterns as desired (Fig. 4). Triangular, square-arrayed and cross-shaped bubble arrays were achieved (Fig. 4a–c) by designing the proper pillar intervals (*a*) and numbers of pillars on each side (*N*) (Supplementary Fig. 8). Figure 4d–f are examples of combinations of the gathering effect and reverse Ostwald ripening. Bubbles have a priority of evolving towards polygonal cells where they can have a larger surface curvature radius until filling them. More generally, complicated patterns were realized, such as peace-dove-shaped bubble arrays (Fig. 4g; Supplementary Fig. 9) and conformal 2D foam with rotational or radial symmetry[26] (Fig. 4h; Supplementary Fig. 10).

**Patterned bubbles serving as a template for assembly of AgNPs.** The properties and applications of the single bubble have been widely studied[27,28]. Meanwhile, a bubble template has proven to be a powerful and promising tool for preparing porous materials including hollow spheres[29], 1D tubular structures[30], 2D mesh structures[31] and 3D metal foams[32]. However, most prepared porous structures are inhomogeneous because of a lack in effective control of the bubbles[33], which results in the nonuniformity of their properties. Here, AgNPs were first assembled because they were used for preparing 2D foams and stayed in the solution. If further evaporation was allowed after forming the hexagonal bubble arrays (Fig. 5a), the boundaries would become narrower and narrower and provided a gradually reduced confined space for the AgNPs

(Fig. 5b). When the liquid completely evaporated, a hexagonal network of close-packed AgNPs forms on the pillar-structured silicon substrate (Fig. 5c). Finally, removing the pillar-structured substrate, an AgNP hexagonal network could be generated on the glass plate (Fig. 5d; Supplementary Fig. 11). SEM and atomic force microscopy were used to image the feature of the AgNPs assembly (Supplementary Fig. 12). The length and width values of the hexagonal sides are $41 \pm 2 \,\mu m$ and $465 \pm 65 \,nm$, respectively. The cross-sectional profile of the AgNP line shows a triangle shape and a height of ca. 200 nm. AgNP networks with different line widths from $48 \pm 13$ to $998 \pm 100 \,nm$, side length from 40 to 160 μm, and various patterns were prepared by varying the concentration of AgNPs and the micropatterned substrate (Supplementary Figs 13 and 14, Fig. 5e–g). For application in electronics, we measured their conductivity and transparency. The data is compared in Supplementary Fig. 15 and summarized in Fig. 5h. The transparency is between 86 and 96%, while the sheet resistance varies from 5.4 to $460 \,\Omega \, sq^{-1}$. In most cases, they outperform conventional ITO transparent thin films[34]. For practical applications, the robustness of these AgNP patterns should be demonstrated. We carried out an immersion test and tape test for the samples after sintering. The variation of conductivity of the samples is within 5% in the immersion test, showing excellent resistance to washing (Supplementary Fig. 16a). After five tape test, the change in conductivity is within 10%, suggesting good performance against abrasion (Supplementary Fig. 16b). The excellent conductivity, high transparency and the robustness

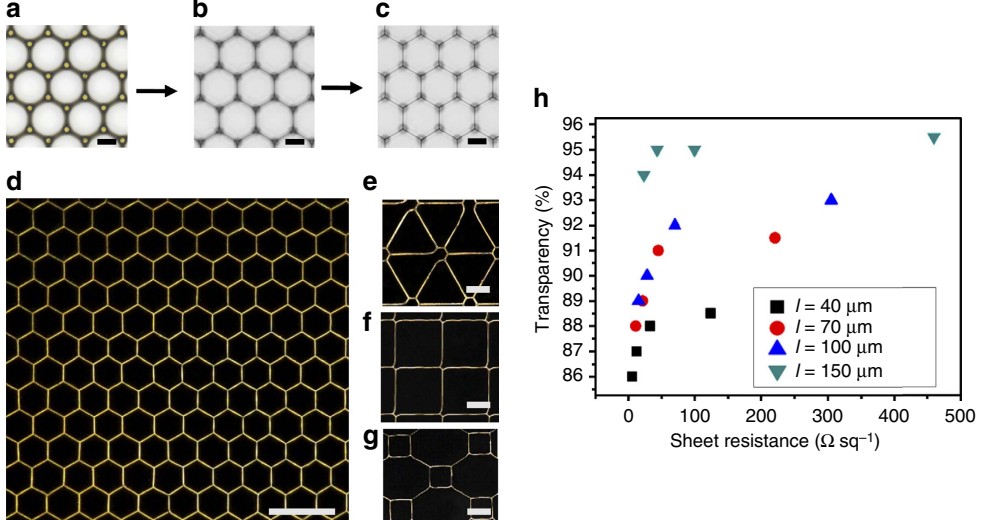

**Figure 5 | Assembling AgNPs into a regular network with the controlled bubbles as a template.** (**a–c**) Microscope observation of assembly process of AgNPs from the top view. As water evaporated, AgNps assembled between the glass and the patterned substrate. (**d**) Dark field optical observation of obtained AgNP hexagonal network after peeling off the glass from the patterned substrate. (**e–g**) Other patterns prepared from corresponding bubble template (Fig. 4a,b,d). (**h**) Sheet resistance and transparency of AgNP hexagonal network (after heat-treatment) depend on the side length, *l*, (from 40 to150 μm) and the line width (from 110 to 1,105 nm) of the hexagonal cell. The spots with the same colour or shape have the equal side lengths and different side widths. Scale bars, **a–c** 30 μm, **d** 100 μm. **e–g** 30 μm.

of the AgNP assembly enable it to be a promising candidate to as an alternative to the expensive ITO. Compared with the promising 3D printed electronic networks[35–37], this method prepares networks with higher precision, shorter preparation time and a simpler setup. In addition, the micropatterned substrate can be recycled by a simple washing process. Therefore as an efficient, clean and sustainable strategy to assembly nanoparticles with nanoscale resolution, it can be tailored for fabricating electronic devices with high-precision.

**Other foaming method to assemble functional materials**. To broaden the applications of this strategy, we have controlled the evolution of 2D foams that are produced by other methods. For example, hydrolysis of sodium borohydride can be catalysed by hydrogen ions to produce hydrogen foams. As shown in Supplementary Fig. 17, the evolution of 2D hydrogen foams is similar to that of 2D oxygen foams produced by the decomposition of urea peroxide. In addition, the conductive polymer, poly(3,4-ethylendioxythiophene) (PEDOT) was assembled with the patterned hydrogen bubbles as the template. The assembly process is almost the same with that of AgNPs with the patterned oxygen bubbles as the template. To broaden the application to lager nanoparticle, we also successfully assembled polystyrene (PS) microspheres with an average size of 450 nm into 2D networks (Supplementary Fig. 18).

## Discussion

These results can stimulate studies in other fields. The patterned bubbles may show different acoustic adsorption corresponding to their structures, which will benefit the research and development of phononic crystals[38,39]. In addition, this strategy beyond Ostwald ripening may be extended to other systems of curvature-driven evolution. We have demonstrated that evolution of 2D liquid foams can be effectively manipulated with a micropatterned surface. The narrow space between micropillars endows the evolution with reverse Ostwald ripening mechanism that weakens the growth of large bubbles.

The special arrangement of pillars contributes the gathering effect which governs the direction of gas diffusion for localizing bubbles. This fundamental understanding guides us to prepare various 2D foam patterns. In addition, with these patterned foams as templates, functional materials from polymers to nanoparticles were assembled. This is expected to be a general strategy to assemble functional materials into desired 2D networks for various device applications.

## Methods

**Fabrication of micropatterned substrates.** Pillar-structured silicon substrates were prepared as previously reported[40]. Silicon wafers (10 cm diameter, N doped, <100> oriented, 525 μm thick) were structured by direct a laser writing apparatus (DWL200, Heidelberg Instruments Mikrotechnik, Germany) that transferred the computer-predefined design onto the wafer coated with photoresist (Shipley Microposit S1800 series) with ~1 μm precision. After irradiation and development, the wafers were etched using deep reactive ion etching (DRIE, Alcatel 601E) with fluorine-based reagents, for different times (10 s to 6 min) depending on the desired height of the structures. Micropillar-structured silicon substrates with tunable arrangements of pillars and pillar intervals were fabricated. After resist stripping (Shipley Microposit Remover 1,165), the substrates were cleaned with ethanol and acetone. To increase the wettability of the surface, the substrates were treated with an air-plasma generator (DT-02S, Ops plasma technology Co., Ltd.) at 200 W for 180 s.

**Preparation of patterned oxygen bubbles and assembly of AgNPs.** AgNPs with an average size of 30 nm were prepared according to the literature[41,42] (Supplementary Fig. 19). In the preparation of patterned bubbles, a micropillar-structured silicon substrate (1 cm² in size) with pillar radius of 5 μm and height of 20 μm, arranged in a designed pattern was held horizontally and then 2 μl of an ethanol solution of urea peroxide was carefully dropped on the substrate and evaporated completely at 4 °C. Some AgNP/(sodium dodecyl sulfonate) SDS suspension (about 0.2%) was dropped on a glass coverslip. The coverslip was placed onto the prepared substrate, yielding a space-confined foaming system. As the decomposition of $H_2O_2$ catalysed by the AgNPs occurred, polydisperse 2D foams were generated. The temperature was kept at 20 °C while the foams evolved. To maintain bubble patterns for a long time, evaporation from the side edges could be avoided by designing a wall surrounding the pillars on the substrate (Supplementary Fig. 20). For assembling AgNPs into a regular network structure, evaporation from the side edges should be allowed. In this case, boundaries among bubbles became narrower and provided a gradually reduced confined space for the AgNPs. Finally, after the liquid completely evaporated, the pillar-structured substrate was removed by physical peeling, and the AgNP networks remained

on the glass substrate. Then, the prepared AgNP networks were immersed in an ethanol solution for 2 h to remove SDS and urea. Finally, the assembly of AgNPs was sintered at 200 °C for 1 h for conductivity measurements.

**Patterning hydrogen bubbles and assembling PEDOT and PS microspheres.** In the preparation of patterned hydrogen bubbles, 2 µl ethanol solution of sodium borohydride was carefully dropped on the substrate and evaporated completely at room temperature. The PEDOT/PSS (Heraeus CLEVIOS PH1000) aqueous solution with 0.2% SDS was dropped on a glass coverslip and put the coverslip onto the prepared substrate, yielding a space-confined foaming system. The hydrogen ion dissociated from PSS (poly(styrene sulfonic acid)) can catalyse the hydrolysis of sodium borohydride to generate 2D hydrogen foams. Because evaporation from the side edges was allowed, boundaries among bubbles became narrower and narrower, which provided a gradually reduced confined space for the polymer. Finally, after the liquid had completely evaporated, the pillar-structured substrate was removed by physical peeling, and the PEDOT/PSS networks remained on the glass substrate. The assembly of PS microspheres with green fluorescence (Shanghai huge biotechnology Co. Ltd.) was similar to that of PEDOT/PSS. The PS microspheres with an average size of 450 nm can be excited at 488 nm and emit at 525 nm. For generating 2D foams, the pH of the PS microspheres dispersion was adjusted to 2 by an HCl solution. A total of 0.2% SDS was added to decrease the surface tension.

**Characterization.** The structures of the assemblies were investigated by scanning electron microscopy (SEM, JEOL, JSM-7500F, Japan) at an accelerating voltage of 5.0 kV. The height map of the AgNP assembly was characterized by an AFM measurement (Seiko SPI 3800N, at a scanning rate of 1Hz in dynamic contact mode). The electrical measurement for the AgNP assembly was characterized by a four-point probe resistivity measurement system (RTS-9, Four Probes Tech Ltd., China). The optical transparency was measured with an angular-resolution spectrometer at a wavelength of 550 nm (Ideaoptics Instruments Ltd., China). The optical and fluorescent images of bubble patterns and the evolution of 2D foams were acquired by an optical microscope (Vision Engineering Co., UK) that was coupled to a charge-coupled device (CCD) camera. The Surface profiles of the patterned surface were characterized by an optical profiling system (Bruker ContourGT-K, USA).

**Simulation.** A 2D simulation was performed using Matlab (Mathworks, Natick, MA). The preparation and evolution of 2D foams in our experiment can be separated because of their disparate time scales. For example, urea peroxide decomposes completely under the catalysis of AgNPs during few tens of seconds, while the evolution of foams lasts dozens of minutes. Therefore, in the simulation, it can be assumed that 2D foams evolve with a constant gas volume fraction. We produced round bubbles with random radii at a given gas volume fraction, on the condition that at the initial state, area of bubbles was less than the cell where they located and the bubbles were distributed evenly. This could be achieved in the experiment by controlling the concentration of urea peroxide and making the urea peroxide uniformly distribute on the patterned substrate. The pillar arrangements in the simulation were identically to those in the experiments. Bubbles evolved with the guidance of pillars under the control of equations (2 and 3) in the main text. Equation (2) was solved by a finite difference method, and the total gas volume was kept conservative in a method described by Lemlich[8]. For the parameters in Supplementary Movie 3, $\Delta t = 0.5$ s, $K = 2$ m$^3$ s$^{-1}$, $a = 35$ µm, and gas volume fraction $w = 57\%$. Moreover, the diameter of the pillars (10 µm) was considered in the simulation to be closer to the real experiment.

Simulation agreed well with the experimental results (Supplementary Movies 2 and 3), except for the oscillations of bubbles after they filled the dodecagonal cells in the simulation movie. The difference could come from the approximation of the theory, which assumed that the concentration of gas in the liquid was equal everywhere; actually, the system needed time to reach an equilibrium of gas concentration in the liquid while the foam was evolving.

**Data availability.** The data that support the findings of this study are available from the corresponding author on reasonable request.

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

## Acknowledgements

Y. Song is grateful for the financial support of 973 Program (Nos. 2013CB933004), the National Nature Science Foundation (grant nos. 51473173 and 51473172), and the 'Strategic Priority Research Program' of Chinese Academy of Sciences (Grant No. XDA09020000).

## Author contributions

Y. Song conceived and designed the experiments. Z. Huang, M. Su, S. Chen, Y. Li and X. Zhou performed the experiments. Q. Yang, Z. Huang and Z. Li ran the simulations. Z. Huang, F. Li and Y. Song analysed the data. Z. Huang, F. Li and Y. Song wrote the manuscript.

## Additional information

**Competing financial interests:** The authors declare no competing financial interests.

**Publisher's note**: 

