## [Peer Review File · Nature Communications]

Reviewers' comments:

Reviewer #1 (Remarks to the Author):

This paper describes the evolution of bubbles within a liquid foam that is confined between a micropillar substrate and a glass plate. While the observations are interesting and potentially useful, I find the paper needs some major revisions to make it to more appeal to the general audience of Nature Communications.

1. The paper is not well written. There are numerous grammar errors; the language is difficult to understand. The explanations are not clear. For example, the authors used both "curvatures" and "curvature radius" frequently and sometimes in adjacent sentences, making the paper hard to read/understand. I suggest authors use consistent terms.
2. It is not clear how the "reverse jungle law" and "defect forming" co-existed. For example, what determines "a"? What does the instantaneous mean radius (ρ) actually mean physically and how/why does it matter?
3. What is the mechanical stability of the nanoparticle assembly? Is this approach generally applicable? To make this practically useful (such as ITO glass as authors tried to compare with), the authors need to demonstrate the robustness of these patterns, for example, against abrasion, washing etc.
4. I am not sure if it is completely correct to call the liquid foams or bubbles two-dimensional. Obviously they are all 3D objects. What did authors mean by 2D? Actually it might be helpful if the authors can take a look at these foams and bubbles in 3D?
5. Again, I appreciate the authors were able to demonstrate a number of different patterns and demonstrate a proof-of-concept application; I just feel the story at its current format may not match the quality and impact of Nature Communications papers. More clear and convincing explanation and demonstration of more broad applications may be helpful.

Reviewer #2 (Remarks to the Author):

Huang and colleagues study the evolution of two-dimensional liquid foams constrained by micropatterned surfaces. Interbubble gas diffusion results in large bubbles absorbing small bubbles, but micropillars arranged in polygonal (or other) shapes are used to limit the maximum radius of gas bubbles. Once the bubble contained by micropillars reaches its maximum circular size, any additional gas flow leads to high-curvature regions (with low radius of curvature) between pillars. The high Laplace pressure of these regions allows for the so-called "reverse jungle law" mechanism whereby smaller bubbles can absorb gas from large constrained bubbles, forming arrays of equal-sized bubbles.

The pillar spacing is identified as a key parameter as this sets the minimum radius of curvature for large bubbles and therefore the radius below which small bubbles will be consumed. Experimental results are shown for hexagonal cells and arrangements containing dodecagonal, hexagonal, and square cells. Simulations are carried out and show qualitative agreement with the observed behavior. Finally, patterned bubbles are used as a template for assembling networks of silver nanoparticles, and conductivity and transparency are compared with conventional ITO transparent thin films.

This work appears to be original and of interest to the community. The experimental and modeling approaches appear sound. However, the manuscript construction does not follow Nature Communications guidelines (e.g. the use of a bold first paragraph in lieu of an abstract, overly terse main text, and relying too heavily on supplemental material). In addition, attention needs to be given to the English grammar and syntax as there are numerous errors that detract from readability. If properly rewritten, the results of this manuscript seem appropriate for publication in a broad-impact journal such as Nature Communications. Below are additional comments and questions:

1) The authors describe the foam evolution in terms of a "jungle law". This choice of words is strange as such evolution is typically referred to as coarsening or Ostwald ripening. Are the authors familiar with other papers within the foam community that refer to this phenomenon as "jungle law", or are there aspects of the phenomenon requiring a new term to be coined?

2) Given the space available, the manuscript would be stronger if the evolution equation were brought into the main text. How do the simulations quantitatively compare with the experiments?

3) Should the reader view these foams as wet or dry? Does this distinction affect the applicability of the model?

4) To what extent do the authors believe the results will extend beyond the specific experimental protocol (urea peroxide solution catalyzed by AgNP)?

5) Fig. 1e shows the phenomenon of "reverse jungle law", but it's not clear in the videos when this process occurs. The submission would be improved if the authors could include a video or longer series of images corresponding to Fig. 1e.

6) The manuscript would be improved if the authors would at least briefly describe the technique for pillar microfabrication in addition to providing the citation. What advantages are there in assembling lattices using two-dimensional foams over more traditional microfabrication techniques?

Other notes:

"Surface curvature radius" should be replaced by "radius of curvature".

Reviewer #3 (Remarks to the Author):

This paper describes a method to develop gas-liquid foam by controlling the surface curvature of the bubbles being generated on a substrate. The effect of pillar spacing of the micro posts and their arrangement is investigated. It is found that the above parameters have a significant effect on the assembly of individual bubbles comprising the foam. The main novelty of this work is that the authors have shown that the bubbles can be guided to evolve in particular shapes by designing the 2D porous media. Furthermore, the authors have shown that their technique can be used to fabricate patterns on glass substrates using Ag nanoparticles.

While the technique to create patterns on glass using 2D controlled foam is a neat application, the reviewer is concerned with the numerous technical inconsistencies in the manuscript. The details of the review are enumerated below.

1. The reviewer disagrees with the usage of the phrases “jungle law” and “reverse jungle law” by the authors in context of these studies. There are plenty of examples in biosphere where small animals may be able to consume larger animals. To make a general sweeping statement with respect to evolution in biological systems, and then to extrapolate it to an engineered system is unacceptable, and appears more like an attempt by authors to make their work look more “attractive” rather than good science. What the authors mean by “jungle law” is essentially the phenomenon of Ostwald ripening where smaller bubbles are generally consumed by larger bubbles. Referring to the phenomenon of Ostwald ripening should have itself been sufficient without resorting to phrases like “jungle law” in this work. Further, the reviewer could not find any precedent in literature where the term “jungle law” has been used even in context of biology.

2. The authors mention that “hitherto there is not an effective way.... electrical or acoustic properties”. The statement is incorrect. Controlling surface curvature is not the only way to manipulate droplet evolution. Use nanoparticles/surfactants/polymers etc. to control surface/interfacial tension in liquids has successfully been done for decades.

3. There is a significant body of literature where foams have been used to fabricate aerogels, metal foams etc. Even the surface curvature of a single bubble has been used as a lithographic tool (see Dressaire, E., et al. (2008). Science 320(5880): 1198-1201 and Yu, G., et al. (2007). Nature

Nanotechnology 2(6): 372-377). The authors should consider citing such previous works. Further, light transmission and sheet resistance measurements of deposited patterns have been performed by many previous authors. The authors should consider citing such works (especially on 3D printed networks by Prof. Jennifer Lewis) which are related to authors experiments in Figure 4.

4. Line 52 and 53: The Laplace pressure is also governed by the interfacial tension and not just the surface curvature.

5. Line 72: The authors talk about the defects that form when the pillar interval changes from 35 μm to 60 μm . First, the meaning of “defects” becomes clear later in the text. Second, the explanation provided by the authors in this paragraph is unclear. The reviewer imagines that with different channel height etc., the pillar spacing at which the “defects” form may also change. So, a more basic question to ask is: what determines the length scales at which these defects occur?

6. The transfer of gas from one bubble to a nearby bubble depends upon the local radius of curvature of each bubble. In case of Fig. 1e for example, the curvature ($k=1/r$) of the bubble at the center of the pattern was smaller than the local curvature of nearby larger bubbles. As a result, the gas diffusion occurred from larger bubble to the smaller bubble. On the other hand, in Fig. 1f, the curvature of the spherical bubble at the center was larger than the local curvature of larger bubbles and hence the diffusion occurred from the smaller bubble to the larger bubble. So, the important quantity here is the value of the local curvature of the bubble. These aspects have been mentioned by the author in their own wordings – but the explanation is unclear, and should be rewritten to make it unambiguous.

7. The experiments represented in Figure 2 are very interesting. However, the logic behind why the authors fabricated these special patterns is unclear. Arguably there could have been many different patterns that the authors could have studied – square, triangle, rectangle etc. The reviewers could have chosen the same pattern but varied the pillar spacing. What guided the authors to choose the special patterns for these experiments?

8. Line 114-115: The authors state “When the total gas is enough, bubbles in hexagonal cells can survive because consumption of bubbles in square cells has filled the dodecagonal cells.” What criterion can be used to know if the “gas is enough”? And how can we make sure that these conditions are met?

9. The universal applicability of the technique is unclear since it requires the fabrication of patterned micro posts first. Further, there are numerous other techniques like 3D printing, solution patterning etc. What advantages are provided by this technique compared to other techniques? In this work, Ag NPs were used – and the bubbles were formed by the catalytic activity of Ag. What if we want to deposit other materials – like aluminum, polymers etc. How can bubbles be produced in such systems, and how can such materials be deposited?

10. Line 161-162: The authors mention “So in most cases, it outperforms the conventional ITO transparent thin films”. First, the authors have given only one example of comparison and not “most cases”. Secondly, they are comparing the performance of their samples with ITO deposited films – which is a different material and hence not a fair comparison. It would be more useful to compare the results with similarly developed patterns through other techniques (such as 3d printing, electrohydrodynamic printing etc.).

11. Towards the end, the authors have mentioned possible applications of their technique can be extended to other systems like water harvesting, anti-icing etc. The implications of Kelvin equation are well-known, but it is unclear how authors work can be used in the application areas that the authors have mentioned. Such type of sweeping statement without significant proof should be avoided.

12. The Methods and Supplementary section, Fig.1 caption: First it is written that ethanol solution of urea peroxide was chosen and a few lines later it is written H₂O₂ is catalyzed by AgNP. There seems to be an inconsistency there.

13. Some references appear on Page 6 and some on Page 8. It is unclear why they are separate.

14. There are numerous typographical and grammatical errors in the text, example

- a. Caption of Figure 1, Line 324: Height is misspelt as hight.
- b. Page 18 of supplementary section: As is the area of the gas/liquid interface which has been typed as as at places.
- c. Page 18 of supplementary section: From is misspelt as form.
- d. Line 41: “Originally, large bubbles...” should be “Initially, large bubbles...”
- e. Line 128: “This reverse jungle law mechanism ensure that....the polygonal cells.”

- f. Line 154: Using “gived” is incorrect grammar.
- g. Caption of Figure 1, Line 331: “The central bubble consumed surrounding larger bubble.” Incorrect tense usage.
- h. Line 70,71: “After that, the growth ability.....the event of “jungle law” does not necessarily happen...”. Improper construct and semantics.
- i. The entire manuscript refers to the pillars endowing surrounding bubbles. Revisiting this sentence construct is strongly recommended.
- j. Line 148/149: “If farther.....”. The word “farther” should be replaced with “further”.
- k. Line 165: “These results can stimulate research in other fields.”
- l. Line 111: “With evolution, bubbles in dodecagonal cells will has a larger surface curvature radius, which causes an unbalanced pressure between dodecagonal cells and square or hexagonal cells.” The word will needs to be deleted.
- m. Supplementary section:
 - “Ethanol solution of urea peroxide was choose because it was easier to spread on the patterned substrate and uniformly distribute than that of hydrogen peroxide water solution.” The tense usage in the italicized words above is incorrect.
 - Page 18: “..surrounding bubbles is difficult because they varies with time and position”. Incorrect tense usage in the italicized word.

REVIEWERS' COMMENTS:

Reviewer #1 (Remarks to the Author):

It is obvious that the authors have done a significant amount of work to address my concerns. I appreciate the efforts. My only remaining concern is that the application part is still relatively weak. It would be helpful if the authors can actually demonstrate some real "device applications" with those additional examples of templated assembly.

Reviewer #2 (Remarks to the Author):

The authors have addressed some concerns with this revision, but two problems still remain.

1. There are still many errors in grammar and sentence structure that detract from readability. Here are some of them with line numbers.

36: "other researches for modeling" should be "other research modeling" or "other studies modeling" (avoid using "researches" to refer to multiple studies, see also 264)

49, 174: "when total gas is not enough and distribute uniformly" is unclear and sounds awkward. (Replace with something like "when the volume fraction of gas is large enough and the gas is distributed uniformly")

74: When bubbles >were< produced in the confined space,

100: "although owning a large size" is awkward

160: To explain that, we calculated quantificationally the radius of curvature - quantificationally is not a word and no adverb is needed there

176: rest bubbles?

199: This is not a quantitative explanation.

252: adaption is not a word (you mean adoption) and the section title is confusing

337: Use a period or semicolon before "for example," because a complete sentence follows it.

This is just a sampling, and the authors should carefully go over the text again to improve the writing. Articles (like "the", "a", "an") are often missing and in some places plural forms should be used where singular forms are and vice versa.

Examples:

99: Then >the< boundary of the bubble

51, 55: "understandings" should almost always be singular

2. The review article you reference in your rebuttal, "Inter-bubble gas diffusion in liquid foam," states in the introduction:

Coarsening (also interchangeably known as Ostwald ripening, inter-bubble gas diffusion or disproportionation) of gas-liquid foam is the process by which big bubbles consume adjacent smaller bubbles due to differences in pressure in the two bubbles caused by the Young-Laplace effect.

There is no need for the new term "jungle law" for a phenomenon that already has multiple names. We believe that sounding impressive is not a good reason to introduce a new term which would add more confusion than insight. Ultimately, whether or not to include the term "jungle law" in the title is up to the editor, but we strongly advise against it (both in the title and the text).

Reviewer #3 (Remarks to the Author):

The author has provided adequate answers to the reviewers questions put forward earlier. Although the reviewer still has reservations on the broader aspects of the study, particularly in context with use of this method to fabricate patterns on surfaces, the technique proposed by the authors has novelty factor that the readers of this journal may find very useful.

One of the first concerns by the reviewer (and by the other reviewers as well) was the use of the term "Jungle Law" by the authors. The authors have done their best to convince that the phrase adequately fits in context of foams as well. The reviewer however still disagrees with the usage of this term. The reviewer agrees that "coarsening" is more widely used term in context of foams, and sees no reason why it could not be used in this work as well. The reviewer disagrees with the notion that usage of "jungle law" phrase will make the work more appealing to the readers, and it should be used because it is "more impressive". Certain degree of flexibility is inherent in science, but it should not be done at the cost of glamorizing phenomenon un-necessarily. Coarsening is a perfectly good word that is used by the scientific community across disciplines in context of foams, colloids, lithography etc. The authors are welcome to frame the term "jungle law" in this work and correlate it with "coarsening" more explicitly.

The authors have made welcome changes in the main text and the supplementary information. However, there are still numerous grammatical and syntax mistakes in the text. The new sentences that have been added by the authors appear misfits sometimes with the rest of the story. The reviewer believes that the manuscript will be appropriate for publication in this journal after the concerns mentioned if the concerns mentioned above are addressed properly by the authors.

Response to the comments of the reviewers.

Reviewer 1

1. Comments: This paper describes the evolution of bubbles within a liquid foam that is confined between a micropillar substrate and a glass plate. While the observations are interesting and potentially useful, I find the paper needs some major revisions to make it to more appeal to the general audience of Nature Communications.

Reply: The authors thank the reviewer for the positive evaluation of our paper. We have revised the manuscript carefully to present our work more clearly.

2. Comments: The paper is not well written. There are numerous grammar errors; the language is difficult to understand. The explanations are not clear. For example, the authors used both "curvatures" and "curvature radius" frequently and sometimes in adjacent sentences, making the paper hard to read/understand. I suggest authors use consistent terms.

Reply: Thanks for the reviewer's comments. We have revised the manuscript to make it easier to read and understand. The grammar errors have been corrected carefully. "Curvatures" and "curvature radius" have been replaced by the consistent term, namely, "radius of curvature".

3. Comments: It is not clear how the "reverse jungle law" and "defect forming" co-existed. For example, what determines "a"? What does the instantaneous mean radius (ρ) actually mean physically and how/why does it matter?

Reply: Thanks for the reviewer's comments. We have added a figure (Fig. 2) to clearly show the situation in which "reverse jungle law" and "defect forming" happen. We will reply the above questions as follows.

1) How did the "reverse jungle law" and "defect forming" coexist?

When bubbles evolve, large bubbles consume small bubbles due to interbubble gas diffusion, but the micropillars arranged in hexagonal shapes limit the maximum radius of gas bubbles. Once the bubble surrounded by micropillars reaches its maximum circular size, any additional gas flow leads to the boundary of bubble deforming into several menisci which have the low radius of curvature and therefore, the high Laplace pressure. **So surrounding smaller bubbles can absorb gas from these large deformed bubbles until having equal radius of curvature. This is the "reverse jungle law",** which prevents the large bubbles infinitely growing and assists to form arrays of equal-sized bubbles.

However, the "reverse jungle law" happens only when the pillar interval, a , is not too large. The pillar interval limit the minimum radius of curvature (namely, $a/2$) for the deformed large bubbles. **Once the deformed large bubble reaches its minimum radius of curvature, the surrounding smaller bubbles with radius above $a/2$ will grow, showing the "reverse jungle law" (Fig. 2a), while bubbles with radius below $a/2$ will shrink (Fig.**

2b), forming a defect. So when a is so large (120 μm , for example) that $a/2$ is larger than the radius of almost all surrounding smaller bubbles, only the defect forms; When a is so small (35 μm) that $a/2$ is smaller than the radius of almost all the surrounding smaller bubbles, only the “reverse jungle law” happens; **When a is medium (60 μm), some surrounding smaller bubbles with radius larger than $a/2$ show the “reverse jungle law” and some with radius smaller than $a/2$ form the defects, so the “reverse jungle law” and “defect forming” coexist.**

- 2) What determines "a"?

The “a” is one of our grammar errors, and we have replaced it with “the”.

- 3) What does the instantaneous mean radius (ρ) actually mean physically and how/why does it matter?

The instantaneous mean radius, ρ , was proposed in Robert Lemlich’s theory (**Please see: *Ind. Eng. Chem. Fundam.* 1978, **17**, 89-93**) for how the foam of arbitrary bubble size distribution might evolve. For a foam with bubbles of different size, the large bubbles grow and the small bubbles shrink until vanishing. **In the system, a critical radius was denoted. If radii of bubbles above it the bubbles will grow, and below it they will shrink. The instantaneous mean radius is identical to the critical radius, physically.**

In Lemlich’s theory, bubbles were viewed as 3D spheres, and ρ was derived as

$$\rho = \frac{\sum_{i=1}^n r_i^2}{\sum_{i=1}^n r_i} \quad (1)$$

Where n is the number of bubbles in the foams, r is the radius of the bubbles. Equation (1) suggests, the instantaneous mean radius was related to all the bubbles in the foam, and it usually increases with time because of the evolution of bubbles.

The instantaneous mean radius matters a lot for predicting the bubble size distribution at any time, as Lemlich deduced,

$$\frac{dr}{dt} = K\left(\frac{1}{\rho} - \frac{1}{r}\right) \quad (2)$$

Where K can be viewed as a constant, t is time. The equation suggests, at any time bubbles with radius larger than ρ will grow and that with radius smaller than ρ will shrink. Besides, the equation can be converted to a finite difference equation that can be easily solved. Therefore it can predict the bubble size distribution of the foams at any time.

In our work, Lemlich’s theory was applied into evolution of 2D foams where bubbles were considered as cylinders instead of spheres. Therefore ρ and evolution equation can be derived as

$$\rho = \frac{\sum_{i=1}^n L_i}{\sum_{i=1}^n \frac{L_i}{r_i}} \quad (3)$$

$$\frac{dA}{dt} = KL \left(\frac{1}{\rho} - \frac{1}{r} \right) \quad (4)$$

Where L , r is the boundary perimeter and radius of curvature of the bubble in 2D, respectively. These two equations are governing equations for the evolution of 2D foams in our simulation.

The instantaneous mean radius also guides us to design the pillar interval for preventing the defect forming. As talked in comment 2, the pillar interval limits the minimum radius of curvature ($a/2$) for the deformed large bubbles. If $r = a/2 < \rho$, according to equation (4), the deformed larger bubbles can be adsorbed and show the “reverse jungle law”. Since ρ gradually increase during the evolution, the minimum ρ is at the initial state of the evolution (denoted as ρ_0). **So keeping that $a/2 < \rho_0$, the defect can be effectively forbidden.**

- 4. Comments:** What is the mechanical stability of the nanoparticle assembly? Is this approach generally applicable? To make this practically useful (such as ITO glass as authors tried to compare with), the authors need to demonstrate the robustness of these patterns, for example, against abrasion, washing etc.

Reply: Thanks for the reviewer’s comments. It has proven that the mechanical stability of Ag nanoparticle assembly or Ag nanowire networks will be improved significantly after heat treatment for sintering (**Please see:** *J. Mater. Chem.* 2011, 21(39): 15378-15382; *Nanoscale*, 2014, 6(9): 4812-4818.). Similarly, we carried out the immersion test and tape test for the samples after sintering generated from our experiments. The results are shown in Supplementary Fig. 16 as follows.

Supplementary Figure 16 | The immersion test (a) and tape test (b) of AgNP patterns after sintering at 200°C for 1 h. (a) Three samples were immersed in 10% water solution of NaOH at 60°C, 6% water solution of HCl at 25°C and solvent composed of water, ethanol and acetone at the rate of 1:1:1 at 25°C, respectively. The conductivity was measured in every 30 minutes. **The variation of conductivity is within 5% in the immersion test, showing good resistance to washing.** (b) The tape test was carried out. **The change in conductivity is within 10% after five tape tests, suggesting good performance against abrasion.**

The excellent conductivity and high transparency as well as the robustness of

the AgNP assembly enable it to be a promising candidate to replace the expensive ITO. Besides, as an efficient, clean and sustainable strategy to assembly nanoparticles with nano-resolution, it can be tailored for fabricating electronic devices with high precision.

5. **Comments:** I am not sure if it is completely correct to call the liquid foams or bubbles two-dimensional. Obviously they are all 3D objects. What did authors mean by 2D? Actually it might be helpful if the authors can take a look at these foams and bubbles in 3D?

Reply: Thanks for the reviewer's comments. Actually, the monolayer bubbles constrained between two plates are usually called 2D or quasi-2D in most of the papers within the foam community. (**Please see:** Chapter 2 in *Foams: Structure and Dynamics*. (Oxford University Press, 2013); Chapter 2 in *Foam engineering: fundamentals and applications*. (John Wiley & Sons, 2012); *Current opinion in colloid & interface science*, 2010, 15(5) 341-358; *Eur. J. Phys.* 2004, 25, 429-438). The 3D structure of the foams constrained between two plates has been studied and shown in the figure below. It suggests, the bubbles of different size in the foams have the same height which equals the spacing between the two plates. So the volume and interfacial energy for the bubble i , can be deduced as

$$V_i = A_i \times h$$

$$E_i = L_i \times h \times \gamma$$

Where V_i , E_i , is volume and interfacial energy of bubble i respectively, h is the spacing between two plates, and A_i , L_i are area and boundary perimeter in 2D from top view respectively. **So the area and boundary perimeter of the bubbles can be representative of the volume and interfacial energy. This is why researchers usually called the monolayer bubbles between two plates 2D or quasi-2D.**

In our work, the liquid foams are viewed as 2D foams similar with discussion above, with details in Supplementary Note 2. **It simplifies greatly the simulation by applying Lemlich's theory into 2D foams.**

Left: Oblique view of a foam between two plates. Right: A ferrofluid foam with slightly higher liquid content, illuminated from below.

Ref: page 57 in *Foams: Structure and Dynamics*. (Oxford University Press, 2013)

6. **Comments:** Again, I appreciate the authors were able to demonstrate a number of different patterns and demonstrate a proof-of-concept application; I just feel the

story at its current format may not match the quality and impact of Nature Communications papers. More clear and convincing explanation and demonstration of more broad applications may be helpful.

Reply: Thanks the reviewer for the positive evaluation of our paper. We have revised the manuscript following Nature Communications guidelines. We also have added more clear explanation in the revised manuscript. In this paper, we have demonstrated that the evolution of wet 2D foams can be well manipulated, instead of obeying the conventional jungle law. The manipulation enabled us to prepare various 2D bubble patterns. With these patterned bubbles as a template, AgNPs were assembled into desired 2D networks.

To broaden the applications of this strategy, we have managed to control evolution of 2D foams which were produced by other methods, such as hydrolysis of sodium borohydride catalyzed by hydrogen ion. In addition, with these patterned bubbles as a template, The conductive polymer, PEDOT (poly(3,4-ethylenedioxythiophene)) and polystyrene (PS) microspheres were assembled into hexagonal networks. As shown in Supplementary Fig. 17 and Supplementary Fig. 18, the evolution of 2D foams generated from hydrolysis of NaBH₄ and the assembly of PEDOT/PSS or PS microspheres are the same with the evolution of 2D foams obtained from decomposition of urea peroxide and assembly of AgNPs, respectively. **In this work, materials from polymer (PEDOT) to nanoparticles with size of 450 nm (PS microspheres) were assembled, therefore we believe this is a general strategy for assembling functional materials into desired 2D networks for device applications.**

Supplementary Figure 17 | PEDOT was assembled into hexagonal network with the patterned hydrogen bubbles as the template generated from the hydrolysis of sodium borohydride catalyzed by the PEDOT/ PSS (Poly(styrenesulfonic acid)) dispersion. (a) The evolution of 2D foams generated from hydrolysis of NaBH₄. (b) The assembly of PEDOT/PSS. (c) Bright field microscope images of obtained PEDOT/PSS network. (d-f) SEM images of PEDOT/PSS network. (g) The molecular structural formulas of PEDOT (top) and PSS (bottom). The hydrogen ion dissociated from PSS in water can catalyze the hydrolysis of NaBH₄ for producing hydrogen bubbles. Scale bars, **a 100 μ m, **b-c** 50 μ m, **d** 20 μ m, **e** 200 nm, **f** 500 nm.**

Supplementary Figure 18 | Polystyrene (PS) microspheres (average size of 450 nm and with fluorescent molecular coating on the surface) were assembled with the patterned hydrogen bubbles as the template generated from hydrolysis of NaBH_4 catalyzed by H^+ . (a-d) Fluorescent micrographs showing the assembly process of PS microspheres, which is the same with the assembly of AgNPs with patterned oxygen bubbles as the template. (e) Fluorescent micrographs of prepared PS microsphere hexagonal network. (f-h) SEM images of obtained PS microsphere network. Scale bars, **a-e 50 μm , **f** 20 μm , **g-h** 1 μm .**

Reviewer 2

- 1. Comments:** This work appears to be original and of interest to the community. The experimental and modeling approaches appear sound. However, the manuscript construction does not follow Nature Communications guidelines (e.g. the use of a bold first paragraph in lieu of an abstract, overly terse main text, and relying too heavily on supplemental material). In addition, attention needs to be given to the English grammar and syntax as there are numerous errors that detract from readability. If properly rewritten, the results of this manuscript seem appropriate for publication in a broad-impact journal such as Nature Communications.

Reply: The authors thank the reviewer very much for the positive evaluation of our work. We have revised the manuscript following the guidelines of Nature Communications. The errors in English grammar and syntax have been carefully corrected. We also moved some figures from supplemental material to Fig. 2 in the main text for less dependence on the supplemental material. Some detailed and clear explanation has been added in the main text for presenting our work clearly.

- 2. Comments:** The authors describe the foam evolution in terms of a "jungle law". This choice of words is strange as such evolution is typically referred to as coarsening or Ostwald ripening. Are the authors familiar with other papers within the foam community that refer to this phenomenon as "jungle law", or are there aspects of the phenomenon requiring a new term to be coined?

Reply: Thanks for the review's comments. We use the words "jungle law" to describe the evolution within various systems that is dominated by surface tension and curvature. In these systems, curvature determines some physical quantities: curvature determines Laplace pressure for gas-liquid foams; curvature determines saturated vapor pressure or solubility according to Kelvin equation for droplets or crystal particles. For ubiquitous cellular structures, such as metals, ceramics, the growth of grain walls have a speed proportional to the mean curvature (**Please see:** *J. phys. Condens. Matter* 1992, 4, 1867-1894). The common phenomenon in these systems is that big individuals absorb the adjacent smaller ones. **In other words, only the larger can survive within these system, which is very similar with the jungle law which means that only the stronger can survive in the jungle.**

The definition of Ostwald ripening recommended by IUPAC in 2007 is that "dissolution of small crystals or sol particles and the redeposition of the dissolved species on the surfaces of larger crystals or sol particles" (**Please see:** *Pure Appl. Chem.*, 2007, **79**, 1801-1829). Although Ostwald ripening often refers to the coarsening and recrystallization process in a nearly saturated solution, sometimes it can be used in a wider sense, such as the evolution of droplets or gas-liquid foams (**Please see:** *Current Opinion in Colloid & Interface Science*, 2010, **15**(5), 374-381). But for the evolution of cellular structures, such as grains and froth, usually is called "coarsening" instead of "Ostwald ripening" (**Please see:** *Nature*,

2007, 446(7139), 1053-1055). **So we use a new and consistent term, “jungle law”, to describe the evolution within these systems. Besides, we believe that “jungle law” is more impressive and appealing to the general audience of Nature Communications.**

3. **Comments:** Given the space available, the manuscript would be stronger if the evolution equation were brought into the main text. How do the simulations quantitatively compare with the experiments?

Reply: Thanks for the reviewer’s suggest very much. We have brought the evolution equation into the main text in the revised manuscript. The simulations agree well with the experiments qualitatively. We have simulated evolution of 2D foams with different gas volume fractions, 34.6%, 55.8%, 56.9%, 71% in Supplementary Fig. 7, which also show qualitative agreement with the results in Supplementary Fig. 6. **However, there are some problems for the quantitative simulation:**

- 1) **It is not well to predict the time the total evolution takes.** As the evolution equation deduced by Lemlich shows, $\frac{dr}{dt} = K\left(\frac{1}{\rho} - \frac{1}{r}\right)$ and $K = 2J\gamma RT/P_a$ (**please see:** *Ind. Eng. Chem. Fundam.* 1978, **17**, 89-93), where J is the effective permeability to the gas transfer. The precise calculation of J in Lemlich’s theory is complex and controversial (**Please see:** *Current Opinion in Colloid & Interface Science*, 2010, **15**(5), 374-381), so the precise calculation of K is difficult, and the time of evolution cannot predict precisely. In our simulation, K was defined as $2 \text{ m}^3\text{s}^{-1}$, and what we pay attention to is the process of evolution (gathering effect) and the final bubble patterns.
- 2) **2D foams with high gas volume fractions (more than 71%) is difficult to simulate.** For wet 2D foams with low gas volume fractions, the evolution obey Lemlich’s model; for dry foams with very high gas volume fractions (more than 95%), the evolution obey von Neumann’s law. However, there has never been a quantitative equation describing the evolution of 2D foams with medium gas volume fractions (**Please see:** *J. Phys.: Condens. Matter*, 1992, **4**, 1867-1894, *Current Opinion in Colloid & Interface Science*, 2010, **15**(5): 374-381) As discussed in Supplementary Note 1, we applied Lemlich’s theory into 2D foams, so the lower the gas volume fraction is, the better the simulation quantitatively agree with the experiments.

Although many efforts still need to make for quantitatively simulating the evolution, we think that our simulations show the “reverse jungle law” due to the narrow space between pillars and “gathering effect” because of the special arrangement of pillars, which agrees well with the experiments.

4. **Comments:** Should the reader view these foams as wet or dry? Does this distinction affect the applicability of the model?

Reply: Thanks for the review’s comments. As we have pointed in the revised main text (Page 2, paragraph 3), **in this work we mainly focused on evolution of foams which was usually called wet foams or bubbly liquid in 2D.**

In Lemlich's theory, the bubbles in the foams were viewed as separate spheres as shown in the figure below, and the evolution was usually called Ostwald ripening. We applied Lemlich's theory into 2D foams evolution, and the bubbles were considered as separate circles. In the foams with high volume fraction, the bubbles squeeze each other and their deformed shapes apart from sphere or circle need to be considered. **So the wetter the 2D foams are, the better our model is applicable to the foams.**

The wet foam or bubbly liquid that is applicable to Lemlich's theory. Ref: *Current Opinion in Colloid & Interface Science*, 2010, **15**(5): 374-381.

For the dry foams whose gas volume fraction is usually larger than 95% as shown in the figure below, the evolution obey the von Neumann's law or von Neumann's coarsening (**Please see:** page 100 in *Foams: Structure and Dynamics*. (Oxford University Press, 2013)), namely,

$$\frac{dA_i}{dt} = \frac{\pi}{3} D_{eff} (n_i - 6)$$

Where D_{eff} is the effective diffusion constant and positive, A_i , n_i are the area and number of sides for the bubble i , and t is time. **The equation suggests, the rate of increase or decrease of a bubble's area depends only on its number of sides.** A bubble with 6 sides will be invariable in the evolution, and the bubble with sides more or less than 6 will grow and shrink, respectively. **This law for dry foams is very different from Lemlich's theory for wet foams. Therefore our model is not applicable to dry 2D foams.**

Dry foam between two plates, viewed from above (bubbles of several mm). **Ref:** page 56 in *Foams: Structure and Dynamics*. (Oxford University Press, 2013)

5. **Comments:** To what extent do the authors believe the results will extend beyond the specific experimental protocol (urea peroxide solution catalyzed by AgNP)?

Reply: Thanks for the review's comments very much. In this work, we studied the evolution of 2D liquid foams constrained by micropatterned surfaces. The narrow space between micropillars endows the evolution with the "reverse jungle law" mechanism. The special arrangement of pillars contributes the "gather effect". These fundamental understandings enable to prepare various 2D bubble patterns. The patterned bubbles can serve as the template for assembling AgNPs into desired 2D networks for electronic applications. **We believe the results can extend beyond the urea peroxide solution catalyzed by AgNPs.**

We do the simulation regardless of the preparation method for the 2D liquid foams. In the simulation, we produced round bubbles with the random radius (in the range between zero and the maximum circular size that the pillars allow for) and at the random position, on condition that bubbles distribute evenly. **Therefore such foams produced by any method will show the "reverse jungle law" and "gather effect".**

We believe some other chemical reactions which emit gas will show the similar results we presents, for example, oxygen bubbles from ethanol solution of hydrogen peroxide catalyzed by ferric ion, carbon dioxide bubbles from carbonate reacting with acid, hydrogen bubbles from hydrolysis of sodium borohydride catalyzed by hydrogen ion, and so on. **To prove the generality of our results, in the revised manuscript we have added experimental results in Supplementary Fig 17 and Supplementary Fig 18.** The evolution of 2D foams consisting of hydrogen bubbles from hydrolysis of sodium borohydride catalyzed by hydrogen ion was studied, showing the same results with urea peroxide solution catalyzed by AgNPs. **In addition, with these patterned bubbles as a template, The conductive polymer, PEDOT (poly(3,4-ethylenedioxythiophene)) and polystyrene (PS) microspheres were assembled into hexagonal networks.** In this work, materials from polymer (PEDOT) to nanoparticles with average size of 450 nm (PS microspheres) were assembled, **therefore we believe this is a general strategy for assembling functional materials into desired 2D networks for device applications.**

6. **Comments:** Fig. 1e shows the phenomenon of "reverse jungle law", but it's not clear in the videos when this process occurs. The submission would be improved if the authors could include a video or longer series of images corresponding to Fig. 1e.

Reply: Thanks for the reviewer's suggestion very much. We have added a longer series of images corresponding to Fig. 1e in the revised manuscript. Please see the supplementary Fig. 5 as follows.

Supplementary Fig. 5 | The longer series of images corresponding to Fig. 2a for showing the “reverse jungle law” in detail. The red arrows denote gas transfer direction among the central bubble and surrounding bubbles due to difference in radius of curvature (denoted with blue dashed circle). Scale bars, 50 μm .

7. Comments: The manuscript would be improved if the authors would at least briefly describe the technique for pillar microfabrication in addition to providing the citation. What advantages are there in assembling lattices using two-dimensional foams over more traditional microfabrication techniques?

Reply: Thanks for the reviewer’s suggest very much. We have added the related description of the technique for pillar microfabrication in the revised Method Section. **The traditional techniques to generate the networks at nanoscale are standard “top-down” microfabrication techniques such as electron beam lithography (Please see: *Nano lett.* 2012, **12**(6): 3138-3144). They commonly involve a long preparation time, complex setup and high cost.** As a result, the “bottom-up” assembly methods have been investigated, such as by inkjet printing (Please see: *ACS Nano*, 2009, **3**(11), 3537-3542), by drying liquid bridges (Please see: *Phys. Rev. Lett.* 2009, **102**(5), 058303) and using a bubble template (please see: *Langmuir*, 2012, **28**(25), 9298-9302). **Nevertheless, the precision of these assembly methods usually are very low** (tens of micrometers), which cannot afford to fabricate electronic devices with high precision. **Alternatively, we present a method combining the top-down and bottom-up options.** The micropatterned surface was fabricated by the bottom-up method called deep reactive-ion etching, which assists the preparation of patterned bubbles. These bubbles serve as the template for assembling AgNPs with a top-down strategy. **This strategy can assemble AgNPs into desired 2D networks with nanosacle resolution. The fabrication is simple, time-saving and low-cost.** In addition, the micropatterned substrate can be recycled by a simple washing process. **Therefore it is an efficient, clean and sustainable strategy and can be tailored to fabricate high-precision electronic devices.**

8. Comments: "Surface curvature radius" should be replaced by "radius of

curvature".

Reply: Thanks for the reviewer's suggest very much. We have replaced the "surface curvature radius" with "radius of curvature" in the revised manuscript.

Reviewer 3

1. **Comments:** the main novelty of this work is that the authors have shown that the bubbles can be guided to evolve in particular shapes by designing the 2D porous media. Furthermore, the authors have shown that their technique can be used to fabricate patterns on glass substrates using Ag nanoparticles. While the technique to create patterns on glass using 2D controlled foam is a neat application, the reviewer is concerned with the numerous technical inconsistencies in the manuscript.

Reply: Thank the reviewer for the positive evaluation of our paper. We have revised the manuscript carefully to clarify our work more clearly.

2. **Comments:** The reviewer disagrees with the usage of the phrases “jungle law” and “reverse jungle law” by the authors in context of these studies. There are plenty of examples in biosphere where small animals may be able to consume larger animals. To make a general sweeping statement with respect to evolution in biological systems, and then to extrapolate it to an engineered system is unacceptable, and appears more like an attempt by authors to make their work look more “attractive” rather than good science. What the authors mean by “jungle law” is essentially the phenomenon of Ostwald ripening where smaller bubbles are generally consumed by larger bubbles. Referring to the phenomenon of Ostwald ripening should have itself been sufficient without resorting to phrases like “jungle law” in this work. Further, the reviewer could not find any precedent in literature where the term “jungle law” has been used even in context of biology.

Reply: Thanks for the reviewer’s comments. The jungle law usually means that “survival of the strongest”, “survival of the fittest”, “kill or be killed”, “eat or be eaten” and “every man for himself”. The definition in *Oxford English Dictionary* is “the code of survival in jungle life, now usually with reference to the superiority of brute force or self-interest in the struggle for survival.” The extended meaning of jungle law has been used in many fields [**Please see:** Ewins P J, “Jungle law in Thailand's forests”, *New Sci.*, 1989, **124**, 42-46; Mackie J L, “The law of the jungle: moral alternatives and principles of evolution”, *Philosophy*, 1978, **53**(206), 455-464; Bonald Thomas, “Is the ‘Law of the Jungle’ Sustainable for the Internet?”, *INFOCOM 2009*, IEEE. 28-36; Whitfield J. “Bushmeat: the law of the jungle”, *Nature*, 2003, **421**(6918), 8-9; Bala K, “Retail drug prices: the law of the jungle”, *Hainews*, 1998, **100**, 2-4]. The term “jungle law” in the biological system means that “survival of the strongest” or “**only the stronger can survive**”. It doesn’t mean that big animals eat smaller ones. Similarly, the code of survival for bubbles in the foam systems is the larger size, and in other words, **only the bigger can survive. From this point of view, the jungle law can describe the free evolution of bubbles in the gas-liquid foams.** We have revised the manuscript carefully to clarify the similarity of biological system and foams more clearly.

The definition of Ostwald ripening recommended by IUPAC in 2007 is that “dissolution of small crystals or sol particles and the redeposition of the dissolved species on the surfaces of larger crystals or sol particles” (**Please see:** *Pure Appl.*

Chem., 2007, **79**, 1801-1829). Although Ostwald ripening often refers to the coarsening and recrystallization process in a nearly saturated solution, sometimes it can be used in a wider sense, such as the evolution of droplets or gas-liquid foams (**Please see: *Current Opinion in Colloid & Interface Science*, 2010, **15**(5), 374-381**). But for the evolution of cellular structures, such as grains and froth, usually is called “coarsening” instead of “Ostwald ripening” (**Please see: *Nature*, 2007, **446**(7139), 1053-1055**). **So we use a new and consistent term, “jungle law”, to describe the evolution within these systems. Besides, we believe that “jungle law” is more impressive and appealing to the general audience of Nature Communications.**

- 3. Comments:** The authors mention that “hitherto there is not an effective way... electrical or acoustic properties”. The statement is incorrect. Controlling surface curvature is not the only way to manipulate droplet evolution. Use nanoparticles/surfactants/polymers etc. to control surface/interfacial tension in liquids has successfully been done for decades.

Reply: Thanks for the reviewer’s comments. The current research for controlling the evolution of foams has been summarized in the introduction, namely, **“To control the evolution of foams, many strategies have been proposed, such as using external field of temperature¹² or magnetism¹³, foams stabilizers of nanoparticles¹⁴⁻¹⁵, protein¹⁶ or surfactants¹⁷⁻¹⁸, insoluble fluorated gases¹⁹ and so on. However, most of these methods can only slow or stop the evolution/aging of foams by weakening the gas diffusion between bubbles”**. According to the comments, the statement has been replaced by **“Heretofore, manipulating evolution of 2D liquid foams beyond the “jungle law” remains a significant challenge.”**

- 4. Comments:** There is a significant body of literature where foams have been used to fabricate aerogels, metal foams etc. Even the surface curvature of a single bubble has been used as a lithographic tool (see Dressaire, E., et al. (2008). *Science* 320(5880):1198-1201 and Yu, G., et al. (2007). *Nature Nanotechnology* 2(6): 372-377). The authors should consider citing such previous works. Further, light transmission and sheet resistance measurements of deposited patterns have been performed by many previous authors. The authors should consider citing such works (especially on 3D printed networks by Prof. Jennifer Lewis) which are related to author’s experiments in Figure 4.

Reply: Thanks for the reviewers’ suggestions. The suggested literatures have been cited, as shown **ref 27-28, ref 35-37** in the revised manuscript.

- 5. Comments:** Line 52 and 53: The Laplace pressure is also governed by the interfacial tension and not just the surface curvature.

Reply: Thanks for the reviewer’s suggestions very much. Laplace pressure is indeed dominated by interfacial tension and the surface curvature. To present our work more clearly and succinctly, we have deleted the sentence in the revised

manuscript.

6. **Comments:** The authors talk about the defects that form when the pillar interval changes from 35 μm to 60 μm . First, the meaning of “defects” becomes clear later in the text. Second, the explanation provided by the authors in this paragraph is unclear. The reviewer imagines that with different channel height etc., the pillar spacing at which the “defects” form may also change. So, a more basic question to ask is: what determines the length scales at which these defects occur?

Reply: Thanks for the reviewer’s comments. We have revised the explanation in the manuscript carefully to explain the formation of defect clearly. The defects also have been denoted in Supplementary Fig. 4. We have added a figure (Fig. 2) to clearly show the situation in which “reverse jungle law” and “defect forming” happen.

As discussed in the revised manuscript, the pillar interval is the factor determining the formation of defects. It matters a lot because it limits the minimum radius of curvature (namely, $a/2$) for the deformed large bubbles. Once the deformed large bubble reaches its minimum radius of curvature, the surrounding smaller bubbles with radius above $a/2$ will grow, showing the “reverse jungle law” (Fig. 2a), while bubbles with radius below $a/2$ will shrink (Fig. 2b), forming a defect. **Actually, the minimum radius of curvature (determined by pillar interval) determines the length scales at which defects occur.** In order to forbid the formation of defects, $a/2$ should be less than ρ_0 (the instantaneous mean radius at the initial state of the evolution). This conclusion will always work regardless of the height of pillars (channel height).

Actually, with different channel height, the pillar spacing at which the defects form will also change, as the reviewer imagines. **The reason is that the ρ_0 varies with the channel height.** For the same foam, the larger of the channel height is, the smaller radius of the bubble will be, and the smaller ρ_0 will be, therefore more defects may form because $a/2$ would not be less than ρ_0 anymore. In this case, the $a/2$ should be less than the corresponding ρ_0 at this channel height.

7. **Comments:** The transfer of gas from one bubble to a nearby bubble depends upon the local radius of curvature of each bubble. In case of Fig. 1e for example, the curvature ($k=1/r$) of the bubble at the center of the pattern was smaller than the local curvature of nearby larger bubbles. As a result, the gas diffusion occurred from larger bubble to the smaller bubble. On the other hand, in Fig. 1f, the curvature of the spherical bubble at the center was larger than the local curvature of larger bubbles and hence the diffusion occurred from the smaller bubble to the larger bubble. So, the important quantity here is the value of the local curvature of the bubble. These aspects have been mentioned by the author in their own wordings – but the explanation is unclear, and should be rewritten to make it unambiguous.

Reply: Thanks for the reviewer’s comments very much. We have revised our explanation according to the reviewer’s suggestion. **Please see: page 3,**

paragraph 3, line 3-10.

8. **Comments:** The experiments represented in Figure 2 are very interesting. However, the logic behind why the authors fabricated these special patterns is unclear. Arguably there could have been many different patterns that the authors could have studied – square, triangle, rectangle etc. The reviewers could have chosen the same pattern but varied the pillar spacing. What guided the authors to choose the special patterns for these experiments?

Reply: Thank the reviewer for the positive evaluation of our paper. We have revised this part in the manuscript for clearly showing our work. In this paper, we investigated the effect of the pillar interval on the evolution. **The same pattern with different pillar spacing has been studied in the preparation of hexagonal bubble arrays of different size, as shown in the revised Fig. 2c-e.** The pillar spacing was demonstrated to determine the “reverse jungle law”.

For studying the effect of arrangement of the pillars. There are various choices, for example Fig. 4d, Fig 4e, Fig 4g and Fig 4h. We choose the pattern of Fig 4f to start our discussion with three reasons as follow.

- 1) **The dodecagonal cell have the much larger area than that of the square (11.2 times) or hexagonal cell (4.3 times), which makes the “gathering effect” very obvious during the evolution.**
- 2) **The formed dodecagonal bubble arrays were discrete.** And during the evolution, the bubbles in different dodecagonal cells cannot touch each other. This greatly simplifies the simulation because the deformation of bubbles owing to contact (Fig 4d, Fig 4e) need not to be considered.
- 3) **The pattern is regular.** The changes in area and radius of curvature for the bubbles can be precisely calculated when the bubbles grow and deform in the square, hexagonal and dodecagonal cells. Therefore quantitative explanations of the “gathering effect” are allowed.

9. **Comments:** The authors state “When the total gas is enough, bubbles in hexagonal cells can survive because consumption of bubbles in square cells has filled the dodecagonal cells.” What criterion can be used to know if the “gas is enough”? And how can we make sure that these conditions are met?

Reply: Thanks for the reviewer’s comments. The simulations of evolution for 2D foams with various gas volume fractions were shown in Supplementary Fig 7. When the gas volume fraction is less than 55.8% (such as 34.6%), the bubbles only fill part of dodecagonal cells. if the volume fraction is more than 56.9% (such as 71% in the simulation), the bubbles in hexagonal cells can survive. **So the “enough gas” in the main text means that the gas volume fraction was about more than 57%.** We have revised the manuscript to clarity it clearly.

10. **Comments:** The universal applicability of the technique is unclear since it requires the fabrication of patterned micro posts first. Further, there are numerous other techniques like 3D printing, solution patterning etc. What advantages are

provided by this technique compared to other techniques? In this work, Ag NPs were used – and the bubbles were formed by the catalytic activity of Ag. What if we want to deposit other materials – like aluminum, polymers etc. How can bubbles be produced in such systems, and how can such materials be deposited?

Reply: Thanks for the reviewer’s comments very much.

1) What advantages are provided by this technique compared to other techniques?

The traditional techniques to generate the networks at nanoscale are standard “top-down” microfabrication techniques such as electron beam lithography (Please see: *Nano Lett.* 2012, **12(6), 3138-3144). They commonly involve a long preparation time, complex setup and high cost. As a result, the “bottom-up” assembly methods have been investigated, such as by inkjet printing (Please see: *ACS Nano*, 2009, **3**(11), 3537-3542), by drying liquid bridges (Please see: *Physical review letters*, 2009, **102**(5), 058303) and using a bubble template (please see: *Langmuir*, 2012, **28**(25), 9298-9302). Nevertheless, the precision of these assembly methods usually are very low (tens of micrometers), which cannot afford to fabricate electronic devices with high precision. Alternatively, we present a method combining the top-down and bottom-up options. The micropatterned surface was fabricated by the bottom-up method called deep reactive-ion etching, which assists the preparation of patterned bubbles. These bubbles serve as the template for assembling AgNPs with a top-down strategy. This strategy can assemble AgNPs into desired 2D networks with nanoscale resolution. The fabrication is simple, time-saving and low-cost. In addition, the micropatterned substrate can be recycled by a simple washing process. Therefore it is an efficient, clean and sustainable strategy and can be tailored to fabricate high-precision electronic devices.**

2) In this work, Ag NPs were used – and the bubbles were formed by the catalytic activity of Ag. What if we want to deposit other materials – like aluminum, polymers etc. How can bubbles be produced in such systems, and how can such materials be deposited?

We believe some other chemical reactions which emit gas will show the similar results we presents, for example, oxygen bubbles from ethanol solution of hydrogen peroxide catalyzed by ferric ion, carbon dioxide bubbles from carbonate reacting with acid, hydrogen bubbles from hydrolysis of sodium borohydride catalyzed by hydrogen ion, and so on. To prove the generality of our results, in the revised manuscript we have added experimental results in Supplementary Fig 17 and Supplementary Fig 18. The evolution of 2D foams consisting of hydrogen bubbles from hydrolysis of sodium borohydride catalyzed by hydrogen ion was studied, showing the same results with urea peroxide solution catalyzed by AgNPs. In addition, with these patterned bubbles as a template, The conductive polymer, PEDOT (poly(3,4-ethylenedioxythiophene)) and polystyrene (PS) microspheres were assembled into hexagonal networks. In this work, materials from

polymer (PEDOT) to nanoparticles with average size of 450 nm (PS microspheres) were assembled, **therefore we believe this is a general strategy for assembling functional materials into desired 2D networks for device applications.**

11. Comments: The authors mention “So in most cases, it outperforms the conventional ITO transparent thin films”. First, the authors have given only one example of comparison and not “most cases”. Secondly, they are comparing the performance of their samples with ITO deposited films – which is a different material and hence not a fair comparison. It would be more useful to compare the results with similarly developed patterns through other techniques (such as 3d printing, electrohydrodynamic printing etc.).

Reply: Thanks for the reviewer’s comments. In the experiments, the transparency and conductivity for a dozen of samples were measured and compared, as shown in the revised Fig 5h. Their transparency is between 86% and 96%, and sheet resistance varies from 5.4 to 460 Ωsq^{-1} . Most of them outperform the ITO thin films. Moreover, “it outperforms” have been corrected as “they outperform” in the revised manuscript.

For the application in transparent conducting films, we measured conductivity and transparency for the AgNP patterns. We compared the performance of our samples with the ITO thin films because ITO thin film has been widely used in transparent conducting films and commercialized. The candidate to replace the brittle and expensive ITO has been widely studied, and they usually compare the performance of their samples with ITO films [Please see: *ACS Nano*, 2009, **3**(11), 3537-3542; *J. Mater. Chem.*, 2011, **21**(39), 15378-15382; *Nanoscale*, 2014, **6**(9), 4812-4818].

According to the reviewer’s suggestion, we also compared our method with the promising and excellent 3D printed electronic networks [**Please see:** *Science* 2009, **323**(5921), 1590-1593; *Nat. Mater.* 2003, **2**, 265-271; *Nat. Mater.* 2016, **15**, 413-418.] in the revised manuscript.

12. Comments: Towards the end, the authors have mentioned possible applications of their technique can be extended to other systems like water harvesting, anti-icing etc. The implications of Kelvin equation are well-known, but it is unclear how authors work can be used in the application areas that the authors have mentioned. Such type of sweeping statement without significant proof should be avoided.

Reply: Thanks for the reviewer’s suggestions very much. We have deleted the statement in the revised manuscript.

13. Comments: The Methods and Supplementary section, Fig.1 caption: First it is written that ethanol solution of urea peroxide was chosen and a few lines later it is written H₂O₂ is catalyzed by AgNP. There seems to be an inconsistency there.

Reply: Thanks for the reviewer’s comments very much. We have corrected the error, and the word “H₂O₂” has been replaced with “urea peroxide”.

14. Comments: Some references appear on Page 6 and some on Page 8. It is unclear why they are separate.

Reply: Thanks for the reviewer's comments very much. We have revised the manuscript, and the references in different sections have been merged.

15. Comments: There are numerous typographical and grammatical errors in the text, example...

Reply: Thanks for the reviewer's comments very much. We have checked the text carefully and corrected the typographical and grammatical errors in the revised manuscript.

Response to the comments of the reviewers.

Reviewer 1

Comments: It is obvious that the authors have done a significant amount of work to address my concerns. I appreciate the efforts. My only remaining concern is that the application part is still relatively weak. It would be helpful if the authors can actually demonstrate some real "device applications" with those additional examples of templated assembly.

Reply: Thanks the reviewer's for the positive evaluation of our work. According to suggestions from the editor, we will put the work about the real "device applications" in another paper.

Reviewer 2

1. **Comments:** The authors have addressed some concerns with this revision, but two problems still remain. There are still many errors in grammar and sentence structure that detract from readability.

Reply: Thanks for the reviewer's comments. We have revised the manuscript about the sentences the reviewer mentioned. The manuscript has been revised carefully to present our work clearly. We also used an English language editing service to improve the clarity and readability of our manuscript.

2. **Comments:** There is no need for the new term "jungle law" for a phenomenon that already has multiple names. We believe that sounding impressive is not a good reason to introduce a new term which would add more confusion than insight. Ultimately, whether or not to include the term "jungle law" in the title is up to the editor, but we strongly advise against it (both in the title and the text).

Reply: Thanks for the reviewer's comments. We have removed the phrase "jungle law" both in the title and the text. The phrase has been replaced with "Ostwald ripening" or "coarsening".

Reviewer 3

1. **Comments:** The author has provided adequate answers to the reviewer's questions put forward earlier. Although the reviewer still has reservations on the broader aspects of the study, particularly in context with use of this method to fabricate patterns on surfaces, the technique proposed by the authors has novelty factor that the readers of this journal may find very useful.

Reply: Thanks the reviewer's for the positive evaluation of our work.

2. **Comments:** One of the first concerns by the reviewer (and by the other reviewers as well) was the use of the term "Jungle Law" by the authors. The authors have done their best to convince that the phrase adequately fits in context of foams as well. The reviewer however still disagrees with the usage of this term.

Reply: Thanks for the reviewer's comments. We have removed the phrase "jungle law" both in the title and the text. The phrase has been replaced with "Ostwald

ripening” or “coarsening”.

3. **Comments:** The authors have made welcome changes in the main text and the supplementary information. However, there are still numerous grammatical and syntax mistakes in the text. The new sentences that have been added by the authors appear misfits sometimes with the rest of the story.

Reply: Thanks for the reviewer’s comments. The manuscript has been revised carefully to present our work clearly. We have revised the related sentences. We also used an English language editing service to improve the clarity and readability of our manuscript.